# The Polymorphic PolyQ Tail Protein of the Mediator Complex, Med15, Regulates the Variable Response to Diverse Stresses

**DOI:** 10.3390/ijms21051894

**Published:** 2020-03-10

**Authors:** Jennifer E.G. Gallagher, Suk Lan Ser, Michael C. Ayers, Casey Nassif, Amaury Pupo

**Affiliations:** Department of Biology, West Virginia University, 53 Campus Drive, Morgantown, WV 26506, USA; suser@mix.wvu.edu (S.L.S.); mcayers@mix.wvu.edu (M.C.A.); Casey.Nassif@gmail.com (C.N.); amaury.pupo@gmail.com (A.P.)

**Keywords:** Mediator, stress, MCHM, Med15, Snf1, polyQ, protein chaperone, master variator, intrinsically disordered regions, yeast, hydrotrope, transcription factors, Myc tag, inorganic phosphate

## Abstract

The Mediator is composed of multiple subunits conserved from yeast to humans and plays a central role in transcription. The tail components are not required for basal transcription but are required for responses to different stresses. While some stresses are familiar, such as heat, desiccation, and starvation, others are exotic, yet yeast can elicit a successful stress response. 4-Methylcyclohexane methanol (MCHM) is a hydrotrope that induces growth arrest in yeast. We found that a naturally occurring variation in the Med15 allele, a component of the Mediator tail, altered the stress response to many chemicals in addition to MCHM. Med15 contains two polyglutamine repeats (polyQ) of variable lengths that change the gene expression of diverse pathways. The Med15 protein existed in multiple isoforms and its stability was dependent on Ydj1, a protein chaperone. The protein level of Med15 with longer polyQ tracts was lower and turned over faster than the allele with shorter polyQ repeats. MCHM sensitivity via variation of Med15 was regulated by Snf1 in a Myc-tag-dependent manner. Tagging Med15 with Myc altered its function in response to stress. Genetic variation in transcriptional regulators magnified genetic differences in response to environmental changes. These polymorphic control genes were master variators.

## 1. Introduction

Changing the transcriptional landscape is a key step in reorganizing cellular processes in response to stress. RNA polymerase II (pol II) transcription is regulated in a stress-specific manner via multiple post-translational modifications and a host of transcription factors (TFs). These transcription factors do not interact directly with pol II and general transcription factors (GTFs), together called the pre-initiation complex, but rather through a multi-protein complex called the Mediator. The Mediator itself is composed of four modules: the head, middle, tail, and kinase domains (Figure 1A). The head interacts with pol II and GTFs, while the tail interacts with specific TFs (reviewed in Verger et al. [1]). The tail is composed of Med2, Med3 (Pgd1), Med5 (Nut1), Med15 (Gal11), and Med16 (Sin4), and the C-terminal end of Med14 connects the tail with the middle of the Mediator complex (Figure 1B [2,3]). The tail is the most diverged between species and the binding of a TF changes the conformation [4]. The Mediator is essential for regulating the expression of most RNA pol II transcripts [5]. Med15, a component of the Mediator tail complex, directly interacts with various transcription factors and has several known phosphorylations that may regulate its function. In yeast, Med15 regulates Oaf1 (fatty acid level sensor), Pdr1 (a transcription factor that regulates pleiotropic drug response), Ino2 (transcription factor involved in inositol response), and Pho4 (basic helix-loop-helix transcription factor of the Myc-family), to name a few [6]. Pho4 is a regulatory factor involved in phosphate metabolism and activates other phosphate regulatory genes, such as *PHO5*, under low phosphate conditions [7]. Being part of the Myc-family, the DNA binding domain of Pho4 has a sequence similarity with various mammalian transcription factors, including Myc, which recognizes the palindromic sequence 5’-CACGTG-3 of basic helix-loop-helix (bHLH) motifs [8]. Chimeras of Pho4-Gal4 in which the bHLH region of the transcription factor was replaced with c-myc remained fully functional [9].

Overexpression of Med15 causes protein aggregation [12], presumably via the poly glutatmine (polyQ) and poly glutamine and alanine (polyQA) regions, as this region alone aggregates in response to hydrogen peroxide [13]. Overexpression of the first polyQ and polyQA of Med15 reduces cell growth in unstressed cells and salt-exposed yeast but rescues growth in the presence of rapamycin [13]. Full-length Med15 also forms cytosolic foci in yeast exposed to hydrogen peroxide [13]. The pathogenic effects of polyQ proteins were uncovered when the causative mutation for Huntington’s disease was discovered [14]. Huntington’s disease causes progressive neurodegeneration in people who inherit a single copy of HTT with the polyQ expansion, inducing protein aggregation (reviewed in Landles and Bates [15]). Aggregation of polyQ expansion proteins in yeast can be reduced by overexpression of chaperone proteins [16]. Ydj1 is a highly expressed general type I Hsp40 protein (J-type) chaperone that localizes to the mitochondria, cytoplasm, and nucleus. A yeast lacking in the Ydj1 function is sensitive to multiple classes of chemicals [17]. Hsp40 proteins work with Hsp70 to refold misfolded proteins or target them for degradation. They also have roles in translation, translocation across membranes, and conformation changes induced by amyloid fibrils. Overexpression of Ydj1 can cure prions [18]. Prions are a group of proteins that not only aggregate but can also induce the aggregation of natively folded proteins. Prions can cause contagious neurodegenerative diseases in humans and switches in the prion state to provide epigenetic plasticity in phenotypic response to stresses by regulating the enzymatic function [12]. Overlapping the polyQ domains are the intrinsically disordered regions (IDRs) that form fuzzy interactions with TFs, in particular, Gcn4 [11,19,20,21]. An N-terminal fragment of Med15 containing the first polyQ and the polyQA domain will form liquid phase condensates, also known as liquid–liquid phase separation, with Gcn4 at low concentrations in vitro [22]. These condensates are dynamic and behave like a liquid (reviewed in Hahn [23]). A mutant of Gcn4 that does not form liquid droplet (phase separation) condensates with Med15 no longer activates transcription [22]. The transition from the single phase to the liquid phase droplet increases the local concentration of factors by forming non-membrane bound compartments that flow and fuse due to surface tension (reviewed in Alberti et al. [24]). Liquid-phase droplets can be induced by chemicals and act as protein concentrators. IDR interactions may be a more general mechanism to increase the local concentration of proteins within liquid droplets, changing protein conformations, and adding complexity-regulating cellular metabolism and environmental responses.

In 2014, there was a large spill of MCHM (4-methylcyclohexane methanol), a coal-cleaning chemical, into the Elk River in West Virginia, contaminating the drinking water of 300,000 people [25]. Many of those people suffered from various significant illnesses, including mild skin irritation, as well as respiratory and gastrointestinal symptoms [26]. Hydrotropes increase the solubility of organic compounds by inducing liquid-phase condensates. Currently, hydrotropes are not considered detergents and detergents function at lower concentrations to solubilize compounds. MCHM acts as a hydrotrope in vitro, preventing protein aggregations [27]. In contrast to ATP [28,29,30] and RNA [31], which are easily metabolized by cells, MCHM can serve as a model hydrotrope to study the effect of hydrotropes on biological systems. RNA and ATP can be rapidly turned over, while MCHM is a cyclic hydrocarbon with saturated bonds that are difficult to break. MCHM is an exotic hydrotrope, and yeast would not have been exposed to MCHM. Exposure to MCHM induced growth inhibition in yeast by changing a wide range of biochemical pathways, including ionome [27] and amino acids [32]. The Mediator binds upstream of many genes across pathways, including stress-responsive genes. Numerous studies have explored the role of Med15 via knockouts on microarrays, and later RNAseq. Removing the entire coding region not only removes the function of a protein but also alters the structure of complexes containing that protein. Gene knockouts are rarely found in nature, while indels, copy number variation, and single nucleotide polymorphism (SNPs) are the most common mutations. By assessing the role of naturally variable proteins, the integrity of the Mediator is maintained and the specific function of Med15 can be addressed in response to hydrotropic chemicals, such as MCHM.

AMP kinases regulate ATP levels, and the yeast ortholog is a heterotrimeric complex called SNF1 (reviewed in Hedbacker and Carlson [33]). In glucose limitation or other stresses, Snf1, the catalytically active kinase in the SNF1 complex, is phosphorylated on T210 by Tos3, Eml1, or Sak1, known as upstream activating kinases (USAKs, reviewed in Hedbacker and Carlson [33]). The protein phosphatase 1 (PP1) negatively regulates SNF1 via dephosphorylation of Snf1 by Glc7 in the presence of glucose. Reg1 regulates the activity of PP1, and in a *reg1* mutant, the SNF1 complex is constitutively active because of the lack of dephosphorylation. Snf1 itself forms liquid droplets in the nuclear vacuole junction (reviewed in Simpson-Lavy and Kupiec [34]). Myc, a commonly used epitope tag derived from transcription factor, c-Myc, is phosphorylated by SNF1, the yeast homolog of AMP kinase, in vitro [35] and Snf1 physically associates with the Mediator complex [36]. SNF1 regulates multiple nutrient-sensing pathways and is important for responses to numerous and diverse chemicals.

MCHM is an exotic hydrotrope that changes the solubility and presumably the structure of proteins. The initial goal of this research was to characterize how a single polymorphic protein, Med15, regulates gene expression in response to a hydrotrope. As the altered states of protein conformation/phase (single versus liquid) are coming to light, the highly variable Med15 was further characterized. Polymorphic transcriptional regulators allow for a small genetic variation to have a large impact on the phenotypic variation. A single polymorphism of threonine to isoleucine removed a potential phosphorylation site in Yrr1, a transcription factor, which confers 4NQO sensitivity but has the benefit of increased respiration [37,38]. These polymorphic proteins are termed master variators [37]. MCHM is a hydrotrope that increases protein solubility [27], and a truncated version of Med15, containing polyQI and polyQA, can exist as liquid droplets in vitro with TFs [22]. Master variators magnify the effect of genetic variation on phenotypic plasticity. We used genetic variation of Med15 to induce differences in the cellular response to MCHM, and in the process, we uncovered how a tag of Myc on Med15 altered the function of the Mediator in conjunction with SNF1 to regulate the cellular response to not only MCHM, but to other diverse stressors.

## 2. Results

The growth of yeast with different components of the Mediator complex being knocked out was tested in terms of the response to MCHM (Figure 1C). As the tail directly interacts with the TFs, *med15*, *med16*, and *med5* knockouts were tested, and *med13* from the CDK was chosen because it is on the other side of the complex from the tail. Mutants in *med5*, *med13*, and *med16* grew better in response to MCHM than the parental strain, BY4741, while the growth of the *med15* mutant was inhibited after two days of growth. Med16 and Med5 are at the distal end while Med2 and Med3 are on the other side of Med15 and closer to the middle module. The *med2* and *med15* mutants were also sensitive to MCHM in YPD (rich media) and more so in YM (yeast minimal media with necessary supplements added to cover the auxotrophies, Figure 1D). We tested a *med15*, *med2* double mutant, which suppressed the MCHM sensitivity in YM and enhanced the sensitivity in YPD. MCHM is a very volatile chemical with a narrow dosage range. Hence, there is much plate-to-plate variation and only strains on the same plate can be directly compared to each other.

Med15 has a curious amino acid sequence (Figure 1E and reviewed in Cooper and Fassler [39]). Med15 from a common lab strain S288c is 16% glutamines and 11% asparagines. Across eukaryotic proteomes, the average Q content is 4% and N content is 5% [40]. The proportion of Q/N amino acids is higher than expected based on the distribution of amino acids and thermophilic organisms do not have this category of proteins [40]. Human Med15 is 20% glutamines, compared to the yeast ortholog with 16% glutamines, and is 27% shorter (Appendix A). While 47% of human orthologs tested can complement yeast knockouts, the percentage decreases for transcription factors and other proteins that associate with DNA [41]. Between species, the C-terminal end of Med15 is highly divergent and required for the association with the Mediator complex via the Mediator activation/association domain. MAD is heavily phosphorylated but the exact roles of these phosphorylations have not been determined [42,43,44,45]. Between polyQI and polyQII, and partially overlapping with the polyQA tract, are three ABDs (activator binding domains) regions [20]. The ABDs and KIX domain form fuzzy interactions with TFs [19,21,46]. The unusual structure of Med15 led us to investigate whether Med15 from other strains had genetic variations in the polyQ tracts. Med15 from five genetically diverse yeast had between 12 and 25 Qs in polyQI, and between 18 and 27 Qs in polyQII. The polyQA only differed by one less QA repeat in the RM11 and AWRI1631 Med15 alleles. There were three other non-synonymous SNPs: K98N, A726T, and V944L, using S288c allele numbering. These strains were then tested on increasing concentrations of MCHM in YPD and YM (Figure 1F). There was a mild decrease in growth in the strains in YM at the highest concentration of 1000 ppm MCHM, which is the limit of solubility of MCHM in media, but there was no difference detected across these strains. These strains are prototrophs and were more robust than BY4741 (Appendix A). Growth was slowed at 800 ppm MCHM in YPD, with YJM789 being the most sensitive.

YJM789 and BY4741 were selected for further study because their alleles of Med15 represented the range of variation in polyQ lengths, differences in MCHM resistance, and available genetic markers (Appendix A). Reciprocal hemizygosity assays were carried out. *MED15* was knocked out in haploid parent strains and diploids were selected. Both the *MED15^YJM789^*/Δ and the *MED15^BY^*/Δ diploids were equally sensitive to MCHM (Figure 2A). However, when compared to the homozygous mutant, the hemizygotes were more sensitive, suggesting there is no impact of the different alleles of Med15 on the MCHM response in the context of a hybrid Mediator complex, but Med15 is possibly important as a gene dosage effect regarding the stoichiometry of the Mediator complex.

The gene dosage could mask allelic differences in the diploid hemizygotes; therefore, to control for this, *MED15* was swapped in BY4741 and YJM789 haploid knockouts. *MED15* alleles were cloned from yeast that had their *MED15* alleles tagged with Myc at the chromosomal location with the KanR marker. Both alleles were expressed under their endogenous promoters from a single-copy plasmid. We did find that the Myc tag on Med15 increased the sensitivity of yeast to MCHM when comparing the reciprocal hemizygotes to the untagged strains. The growth of Myc-tagged hemizygous mutants was inhibited at 550 ppm, while the untagged mutants were inhibited at 650 ppm (Figure 2A and Appendix A, respectively). *MED15* was knocked out in BY4741 and YJM789 and transformed with the two alleles of Med15-Myc, with the empty plasmid as the negative control, and were grown in YPD or YM (with glutamate (MSG) as the nitrogen source instead of ammonium sulfate to maintain the selection of the KanR plasmid in minimal media with G418). Wildtype BY4741 grew slower than BY4741 carrying Med15^S288c^-Myc in YPD with low levels of MCHM for two days (Figure 2B, rows 1 and 4). In these same conditions, there was very little change in the growth of the BY4741 *med15* knockout (Figure 2B, row 2). However, the Myc was severely affected by MCHM. Consistent with the growth of the other strains, more MCHM was required in YM to slow the growth of yeast and there was no difference between the three alleles of Med15 (rows 5, 6, and 8). The *med15* knockout grew slower in YM but appeared to be unaffected by MCHM when the slow growth of this mutant was also taken into account (Figure 2B, row 6). YJM789 growth was not affected by the alleles of the Med15-Myc expressed in YPD or YM (Figure 2B, rows 9, 11, 12, 13, 15, and 16). The YJM789 *med15* mutant grew slower in YM, yet the mutant growth was about the same in 400 ppm MCHM in YPD and YM. At 550 and 650 ppm MCHM in YPD, the knockout grew better in YPD than yeast with Med15-Myc compared with the growth in YM.

It was surprising that the BY4741 Med15^YJM789^-Myc yeast was more sensitive to MCHM in YPD than the *med15* knockout yeast. To test whether the Med15^YJM789^-Myc was a dominant-negative allele, Med15^YJM789^-Myc was expressed in wildtype BY4741 with endogenous Med15^BY^. Expressing both Med15^YJM789^-Myc and Med15^BY^ in yeast did not change the growth in YPD with MCHM and no difference was noted when compared to yeast with Med15^S288c^-Myc and Med15^BY^ in the BY4741 *med15* knockout (Appendix A, rows 7 and 8). However, the yeast expressing both Med15^YJM789^-Myc and Med15^BY^ were more sensitive to MCHM in YM with high levels of MCHM (Appendix A, rows 7 and 8). To assess the impact of the variation of each polyQ tract on the yeast growth in the presence of MCHM, the polyQ domains from Med15^S288c^ were used to replace the respective domains in Med15^YJM789^. Both polyQ^S288c^-swapped alleles improved the growth of the yeast containing Med15^YJM789^ (Figure 2C). Although, each polyQ^S288c^ swap partially rescued the growth defect contributed by the Med15^YJM789^ allele, we continued studies with Med15^YJM789^ and Med15^S288c^ for comparison.

To determine whether the protein levels of Med15 contributed to differences in MCHM sensitivity, protein levels of the cloned alleles Med15-Myc in the allele swapped strains were measured. Med15-Myc proteins were immunoprecipitated because the levels were too low to detect using Western blotting without enrichment. The yeast was grown to mid-log phase in YPD or YM supplemented with amino acids and then shifted to media containing MCHM for 30 minutes. Med15^YJM789^-Myc levels were lower than Med15^S288c^-Myc in all conditions tested: YPD, YM, and with and without MCHM. In general, the levels of both alleles were lower in YM. Med15^YJM789^-Myc levels also appeared to decrease in YPD with MCHM but the decreased levels did not explain the MCHM sensitivity as the *med15* knockout was not as sensitive as the yeast carrying the Med15^YJM789^-Myc allele. Similarly, the yeast with Med15^YJM789^-Myc grew similarly to yeast carrying Med15^S288c^-Myc in YM, and the levels of Med15 protein were very different in YM (Figure 3A). It is also curious to note, with the Myc-tag, Med15^S288c^ was predicted to be 140 kDa with a pI at 6.61 and Med15^YJM789^ was predicted to be 142 kDa with a pI at 6.48. The Med15^S288c^-Myc protein ran above the 150 kDa marker as multiple bands despite being shorter than Med15^YJM789^-Myc, which ran truer to size. In part, the differences in Med15 protein levels could be attributed to the differences in mRNA levels. Global mRNA levels were quantified using Illumina sequencing of three biological replicates (Figure 3B). *MED15^YJM789^*-Myc mRNA decreased in YPD with MCHM and was equivalent in YM irrespective of MCHM. The *MED15^S288c^*-Myc mRNA levels also tracked with protein levels. The *MED15* promoter contained four SNPs that were included on the plasmid, which were in relation to the start codon of S288c to YJM789: A-8T, A-209G, A-365G, and T-449C. Next, the stability of the Med15 proteins was measured via treatment with cycloheximide, which blocked translation. While Med15^YJM789^-Myc protein levels were lower than Med15^S288c^-Myc, by the end of the time course, Med15^YJM789^ had decreased more relative to the levels of Med15^S288c^ (Figure 3C).

Med15 is important for the response to many different stresses, and to determine which genes were differentially regulated, RNAseq was carried out. BY4741 and its isogenic *med15* knockout were grown to log-phase and then treated with MCHM. In YPD, 149 genes were up-regulated and 184 genes were down-regulated in the *med15* knockout compared to BY4741 (Figure 4A and Appendix A). The down-regulated genes were related to the metabolic processes of nucleosides and ribonucleosides, pyruvate metabolism, carbohydrates, organophosphates catabolism, small molecule biosynthesis, and oxidoreduction coenzyme metabolism, among others (Appendix A). The down-regulation of these metabolic genes in the absence of the Med15 protein may, in part, explain the reduced growth of the knockouts in rich media or the regulation of these genes themselves were dependent on Med15.. Other conditions that reduce growth, such as petite yeast [27] or treatment with chemicals which reduce growth [38], also demonstrate the down-regulation of similar pathways. In YPD, 46 GO terms were up-regulated and 76 were down-regulated, while in YM, 35 were up-regulated and 72 were down-regulated. This set was not enriched in genes related to the heat-shock response, drug/toxin transport, stress response, or cellular import as in Ansari et al. [47], or in ribosome biogenesis as in Miller et al. [6]. Sporulation-related genes were up-regulated (Appendix A), as previously reported in References [47,48], although they are not yet known to have functional relevance in haploid cells. There were also genes involved in cell development, reproduction, morphogenesis, and sulfur compound biosynthetic processes. Previously, a study found that genes were up-regulated for sulfur metabolism in the *med15* mutant [47].

When MCHM was added, the number of differentially expressed genes increased in the *med15* mutants. There were 468 up-regulated genes and 278 down-regulated genes (Figure 4B and Appendix A). There was extensive overlap in the metabolic pathways in the down-regulated genes in the *med15* knockout compared to BY4741 in YPD only and YPD + MCHM, with only three more GO terms appearing: monosaccharide metabolism, organic acid, and carboxylic acid biosynthesis (Appendix A). The difference was significant in the up-regulated genes, not only in the number, but also in their functionality, as the GO terms overlap was low and a wide set of new terms related to ribosomes, polyamine transport, and RNA export from the nucleus appeared. It is of note that in our study, *med15* deletion caused the up-regulation of ribosome biogenesis genes, contrary to the down-regulation observed in Miller et al. [6]. Furthermore, their observed down-regulation of this set of genes was the same in wildtype versus *med15* under osmotic stress, while we only observed the up-regulation in the presence of MCHM, suggesting a fundamentally different mechanism of responding to osmotic stress and MCHM-induced stress in yeast.

By directly comparing the Med15^YJM789^-Myc and Med15^S288c^-Myc effects on gene expression, Med15^YJM789^-Myc changed the expression of 69 genes and Med15^S288c^-Myc changed 23 genes compared to BY4741, when treated with MCHM in YPD ( Figure 4C,D, respectively, and Appendix A). The functional impact may be minimal as no term came out of the GO analysis (Appendix A). Eight out of the nine down-regulated genes in Med15^YJM789^-Myc versus BY4741 were involved in the small molecule biosynthetic process (Appendix A). Besides ribosome biogenesis, there were up-regulated genes related to rRNA processing, ribonucleoside and glycosyl compound biosynthetic processes, and ion transport (Appendix A).

The change of media (YM instead of YPD) provoked a significant change in gene expression variation among the different cases being compared (Appendix A and Appendix A). However, the functional analysis of down-regulated genes was strikingly similar to that of yeast grown in YPD (Appendix A). The functional analysis of up-regulated genes in YM showed a different picture with *med15* knockout versus BY4741, where GO terms were almost the same regardless of the presence of MCHM, but three new GO terms appeared in Med15^YJM789^-Myc versus BY4741: sulfate assimilation, cysteine biosynthesis, and secondary metabolism.

The Mediator tail module preferentially associates with SAGA-dominated genes [49]. We determined whether the differentially expressed genes from this RNAseq overlapped with the SAGA and TFIID-dominated categories. From the supplementary table where all genes are labeled depending on their SAGA/TFIID dominated status, we analyzed the appropriate annotation for the 1111 unique genes that were differentially expressed in any of the comparisons (Appendix A). From these, 1012 genes were found in the supplementary table [49]. We have added a pie chart with the labeling of our relevant genes (Appendix A). As can be seen, the SAGA-dominated set was a minority at 26%, but given that SAGA-dominated genes are just 10% of the genome [49], then there was an enrichment of those genes (of 2.6× compared to a randomly sampled set of genes of equivalent size). Med15 binds upstream of many genes [50]. Three genes were chosen for further characterization, namely *PTR2*, *PUT4*, and *YDJ1* [51]. Ptr2 is a dipeptide transporter [52] and Put4 is the high-affinity proline permease [53]. Nitrogen catabolite repression down-regulates transporters and permeases of nonpreferred nitrogen sources when preferred nitrogen sources, such as ammonium or glutamine (MSG), are available (reviewed in Mara et al. [54]). At the cell membrane, both of these proteins are down-regulated via endocytosis when shifted to a preferred carbon source, with Put4 degradation being faster than Ptr2 [55]. Except in MCHM treatment in YM, the levels of *PTR2* were significantly decreased and the levels of *PUT4* significantly increased in Med15^YJM789^ compared with Med15^S288c^ in all other conditions, while the levels of *YDJ1* expression remained the same (Figure 5A). The knockouts of these genes conferred MCHM sensitivity in YPD. However, in YM, only the *ydj1* yeast strain was also sensitive to MCHM (Figure 5B). The role of Ydj1, a protein chaperone, on Med15 function was further characterized. MCHM acts as a hydrotrope that alters protein solubility, which is related to protein conformation. Swapping the Med15 alleles in the *ydj1* knockout had no effect on growth. The *ydj1* knockouts were slow-growing in BY4741 (Figure 5B,C) and *ydj1* is lethal in W303 [56]. The impact of the loss of Ydj1 on Med15 protein levels was measured using Western blotting (Figure 5D). The Myc tagged protein isoforms were more heterogeneous in size in the *ydj1* mutant and the levels of Med15^YJM789^ increased to match that of Med15^S288c^. The slowest migrating band of Med15^YJM789^ increased to match that of Med15^S288c^.

Med15 contains multiple phosphorylations with the C-terminal MAD. It is unknown whether these phosphorylations are regulated in a stress-dependent manner. Both alleles of Med15-Myc ran as multiple bands that did not appear to change in the MCHM treatment (Figure 3A). To determine whether other stressors could alter the isoforms of Med15, yeast (BY4741) expressing Med15^S288c^-Myc was treated with either MCHM or hydrogen peroxide over a 90-minute time course. There was no visible change in the pattern of Myc-tagged proteins in the Western blot (Appendix A). While the effects of MCHM on yeast clearly point to an increase in ROS stress for the cell, wildtype cells seemed to be robust enough, on average, to limit this stress, while mutants in certain pathways could not. To test this hypothesis, we also performed the dihydroethidium (DHE) assay on *med15* mutants (Appendix A). The endogenous levels of ROS were higher in *med15*, and when treated with hydrogen peroxide, the ROS increased compared to BY4741. The wildtype and mutant strains showed a similar pattern of DHE fluorescence when treated with MCHM, with the appearance of a high ROS population peak that was less intense and broader than hydrogen peroxide. The major difference from the wildtype was an increase in the size of the high ROS population peak. Therefore, *med15* mutants may have an innate sensitivity to MCHM due to their inability to maintain ROS homeostasis.

The growth analysis, protein levels, and transcriptomics of the Med15 allele swaps were carried out with Myc-tagged alleles ( Figure 2, Figure 3, Figure 4 and Figure 5). Numerous studies have used these epitope tags, and on occasion, have noticed negative effects on the function of the protein. The typical control is testing the growth of yeast. Strains carrying either of the Myc-tagged alleles experienced growth that was indistinguishable from untagged alleles on YPD and YM in BY4741 and YJM789 (Figure 2B). However, we began to question this pattern, at least in some stress conditions, upon deeper analysis of the RNAseq ( Figure 4D and Appendix A). Med15 was among the overexpressed genes in Med15^S288c^-Myc versus BY4741 (log2FC ≈ 1.1) in MCHM treatment and 22 other genes also changed expression (Figure 4D). There was no statistical difference in the expression levels of the tagged allele in untreated YPD or YM (Appendix A). Yeast expressing Med15^S288c^-Myc showed a few genes that were up-regulated in both YPD and YM, such as *PUT4* and *PHO89.* Genes that were down-regulated encoded protein chaperones: *HSP30*, *HSP4*, and *SSA4* (Appendix A). Initial experiments were carried out in the prototrophic S288c (GSY147) to compare it to YJM789 (Appendix A), and the cloned Med15^S288c^-Myc was from genomically tagged GSY147 and YJM789 and inserted into pRS316. However, to take advantage of the Yeast Knockout Collection and study allele effects in a single genetic background, subsequent experiments, including RNAseq, were carried out in BY4741. Med15 polyQII from GSY147 contained 23 glutamines, while BY4741 had 18 glutamines (Figure 1E). We compared the impact of shorter polyQ tracts in Med15 by comparing RNAseq from yeast carrying p*MED15^S288c^*-Myc to BY4741. Genes such as *PHO89*, *PUT4*, *SSA4*, *HSP30*, *URA1*, and *MDH2* were differentially expressed in both YPD and YM (Appendix A). All strains tolerated MCHM better in YM compared to YPD [27], and intracellular levels of metals and other ions increased in MCHM, including phosphate, which doubled in YPD with the MCHM treatment [27]. Knockouts of *pho89* and a related phosphate transporter, *pho84*, were grown in the presence of MCHM. Despite the *PHO89* expression increasing in RNAseq, the knockout grew the same as BY4741 and the *pho84* mutant was sensitive to MCHM in YPD and YM (Appendix A). *PHO89* expression was higher in *med15* knockouts in all conditions tested ( Figure 4A,B and Appendix A), and we concluded that Med15 negatively regulated the expression of *PHO89* independent of MCHM. In the RNAseq, the presence of the Myc tag was not taken into account.

Between Med15^S288c^ and Med15^BY^, only polyQII differed (18Q versus 27Q). To explore whether the variation in the polyQII or the presence of the Myc tag was affecting the MCHM response, Med15 was tagged at its genomic location in BY4741 and compared to BY4741 carrying Med15^S288c^ with and without the Myc tag. The yeast with Med15^S288c^ without the Myc tag grew slower than the yeast carrying Med15^S288c^-Myc. The presence of the Myc tag on Med15^BY^ slightly increased the MCHM tolerance relative to Med15^BY^ but not as much as the yeast with Med15^S288c^-Myc (Appendix A). However, these differences were only seen at a lower concentration of 350 ppm MCHM at day 2 compared to concentrations used for RNAseq or initial screening for three days of growth (Figure 6A). The effect of the Myc tag on Med15 was directly tested by cloning endogenous Med15^YJM789^ without the Myc tag and then testing the growth of yeast on MCHM. The difference in MCHM sensitivity was lost when the Myc tag was absent (Figure 6A). In a genomic screen of the knockout collection for MCHM-sensitive mutants, both *snf1* and *reg1* mutants were identified [57]. Myc is likely phosphorylated by Snf1 in vitro [35]. Reg1 is the regulating subunit of the phosphatase complex that dephosphorylates Snf1 at T210, which down-regulates its kinase activity. In a *reg1* mutant, the SNF1 complex has more kinase activity (reviewed in Hedbacker and Carlson [33]). *snf1* mutants grew slightly slower than wildtype yeast on YPD and were more sensitive than the *med15* mutant to MCHM, while *reg1* mutants grew only a little slower (Figure 6B). The *snf1* + *med15* double mutant growth was between the two single mutants and the *reg1* + *med15* double mutant was closer to the *med15* single mutant. In *snf1* mutants, only expression of Med15^S288c^-Myc affected growth in MCHM (Figure 6C). The pattern of Med15 bands in *snf1* and *reg1* mutants was measured using Western blotting. Both Med15 alleles were similar in the wildtype yeast and *snf1* mutants, but Med15^YJM789^-Myc shifted up and became more similar to Med15^S288c^-Myc in the *reg1* mutant (Figure 6D).

The two alleles of Med15-Myc conferred different phenotypes not only against MHCM, but also against other chemicals (Figure 7A). The yeast was grown in an automatic plate reader and the growth difference maximum between alleles during the log-phase was plotted. The yeast with Med15^YJM789^-Myc had a greater resistance against compounds that directly generated free radicals, such as hydrogen peroxide, and 4NQO, which generates free radicals as it is metabolized [38]. MCHM is a volatile compound [58], and when quantitative growth assays were carried out in small volumes, the MCHM evaporated before the end of the growth assay such that growth was only marginally slower in yeast with Med15^YJM789^-Myc. The Med15^S288c^-Myc allele conferred resistance to reducing agents that cause an unfolded protein response, such as beta-mercaptoethanol and DTT; to DNA damaging chemicals, such as camptothecin; and to hygromycin, which inhibits translation. The yeast with Med15^S288c^-Myc was also more resistant to caffeine, which in part can mimic the effects of TOR inactivation, but not to rapamycin, which also inhibits TOR. Other chemicals that did not differentially inhibit yeast with different Med15 alleles were Credit41 (glyphosate-based herbicide), which inhibits aromatic amino acid biosynthesis, and hydroxyurea, which arrests cells in S phase by depleting nucleotides. From this panel of 13 chemicals, 7 were chosen for further characterization in yeast with different alleles of Med15 with and without the Myc tag (Figure 7B). At times, the Myc tag flipped the preference of the allele and at other times exaggerated the differences. Only in the presence of caffeine and hygromycin was no difference in growth seen between the untagged alleles, yet increased growth was seen of the yeast carrying the Med15^S288c^-Myc allele. In calcofluor white, the Med15^S288c^-Myc yeast grew better, and in camptothecin, yeast with Med15^S288c^-Myc grew better, while the untagged Med15^S288c^ was marginally better than Med15^YJM789^ for both chemicals.

To assess the impact of the loss of Snf1 and Reg1, quantitative growth assays were conducted in allele swaps with *snf1* and *reg1* mutants. *snf1* mutants with untagged Med15^S288c^ had a slightly improved growth compared to *snf1* Med15^YJM789^ in many conditions, including in YPD, but the growth was not as much as wildtype yeast with the same allele (Figure 7C). In contrast, Med15^YJM789^-Myc containing *snf1* yeast grew better in 4NQO, calcofluor white, caffeine, DTT, and hygromycin. Hydrogen peroxide and 4NQO both produced ROS but using different mechanisms. Hydrogen peroxide was directly converted to ROS and 4NQO was converted through a respiration-dependent mechanism [38]. *snf1* yeast with Med15^S288c^-Myc allele grew better in hydrogen peroxide than wildtype yeast with Med15^S288c^-Myc. Th loss of Reg1 had a similar trend in the changes of growth with the notable exception of 4NQO. In that case, the *reg1* Med15^S288c^-Myc grew better than the wildtype yeast (Figure 7D).

## 3. Discussion

The expansion of polyQ in proteins was discovered to be the cause of numerous neurodegenerative diseases. Slippage of the DNA polymerase during DNA replication and unequal homologous recombination causes expansion and contraction of the repeats. In Huntington’s disease, expansions over 30 repeats are considered pathogenic and induce the aggregation of Huntington proteins. In vivo, these aggregates form foci in the cell that under static imaging cannot be distinguished from liquid phase-separated condensates. The function of hydrotropes in biology is recently becoming appreciated, where it regulates the reversible formation of protein condensates. Several proteins in the Mediator complex have IDRs that promote liquid phase-separated condensates with TFs [22]. Intrinsically disordered regions, such as polyQ tracts, facilitate phase separation [11,19,21,22,46,59]. The two polyQ tracts in Med15 vary between 12 and 27 repeats between strains. Changes in the polyQ tracts of Med15 changed the response to numerous chemicals and were dependent on Snf1. Throughout the tail proteins of the Meditator, there is genetic variation that has yet to be explored. The reciprocal hemizygosity of *med15* mutants did not differentiate between the YJM789 or BY4741 alleles. Both hemizygous mutants were more sensitive to MCHM than the homozygous mutant, despite the YJM789 strain having a higher tolerance to MCHM than BY4741. This was the case when Med15 was tagged with Myc. Allele swapping of Med15^YJM789^ into the BY4741 background conferred MCHM sensitivity but only in YPD and when the Myc tag was present. While in YJM789, the expression of Med15^S288c^-Myc did not change the MCHM resistance. In both BY4741 and YJM789 strains, the *med15* mutants were slow-growing in untreated media, which was not affected at higher concentrations of MCHM, making it appear that at the highest concentrations of MCHM, the YJM789 *med15* mutants were resistant to MCHM. MCHM sensitivity induced by the expression of Med15^YJM789^ in BY4741 was not dominant. Therefore, we concluded that the YJM789 Mediator complex could better tolerate Med15 with shorter polyQ tracts, while the BY4741 Mediator was more sensitive to perturbations. Myc possibly increased the recruitment of Snf1 to the Mediator when Med15 contained expanded polyQ tracts. *MED15*^YJM789^ was expressed at slightly lower levels and the protein level was at an even lower level than Med15^S288c^ and was also less stable. While changes in mRNA levels contributed to lower protein levels, the longer polyQ tracts in Med15^YJM789^ may have also slowed the translation or increased the ubiquitin-dependent degradation, as the protein was less stable when the translation was inhibited.

Ydj1 was required for the stability of the Med15 protein and it was difficult to assess the role of Ydj1 on the Med15 protein’s stability because of the extremely slow growth of the *ydj1* mutants. Ydj1 also has a role at H3 histone eviction when transcription is induced. Gcn4 binding to promoters is not reduced in a *yjd1* mutant [60] or at the *GAL1* promoter [61]. Hsp70 associates with several Hsp40-like proteins, including Ydj1, a type 1 Hsp40 that stimulates Hsp70 activity. Ydj1 is localized to the perinuclear and nuclear membranes [56]. The role in nucleosome eviction may be indirect by helping to fold Med15. Ydj1 inhibits the SDS-resistant aggregation of the polyQ containing a section of Htt in yeast [16,62]. The Med15 fragment containing the polyQ aggregates in vivo [13], as well as when full-length Med15 is overexpressed. Ydj1 was required for both alleles of Med15 protein stability as the isomers that were Myc tagged became less distinct such that Med15^YJM789^ remained true to size in contrast with Med15^S288c^. Zinc can aid in catalysis as an enzyme cofactor but also stabilize the structure of proteins when bound; furthermore, Ydj1 binds zinc. In response to zinc starvation, yeast ration their zinc, known as zinc sparing [63]. Intracellular zinc levels are three times higher when treated with MCHM, and supplementation with zinc improves growth up to the point when too much zinc can no longer rescue MCHM-induced growth arrest [27]. The Ydj1 protein level is dependent on zinc levels, although transcription was not affected [63]. For Ydj1, zinc may serve to stabilize the protein structure, protecting it from degradation as it unfolds without zinc. Yjd1 may function to regulate the phase state of Med15 or other components of the Mediator.

Yeast exposed to MCHM up-regulate many pathways involved in biosynthesis; however, these yeast do not appear to be lacking for these nutrients. It was shown that amino acids [32], inositol [32], zinc [27], and phosphate [27] levels are increased. Since the 2014 MCHM spill, several studies have measured the toxicological effects of MCHM on diverse species but have not addressed the mechanism of toxicity during acute exposure [64,65,66,67,68,69,70,71]. One possible explanation of MCHM’s diverse effects is, as a stable hydrotrope, MCHM changes the structure of nutrient sensors so they no longer sense extracellular compounds. In YM, more genes were differentially expressed and yet yeast were more tolerant to MCHM. Compared to YPD, in YM, nutrient transporters are down-regulated, and the biosynthetic pathways are up-regulated, which could mitigate the effect of MCHM. Familiar compounds, such as ATP and RNA, are hydrotropes and induce protein condensates, which are separated from the surrounding proteins in liquid phase separation. This serves to concentrate functional proteins reversibly rather than inactivate them as protein aggregates. MCHM is a cyclic hydrocarbon that is relatively more stable than RNA and ATP in the cell. MCHM was detected in sediment ten months after the spill [71]. MCHM is primarily degraded into aldehydes and carboxylic acids [72], which generates ROS [73]. The increased ROS seen in this study was seen at 12 hours after incubation and therefore the increase in ROS from degradation would be a secondary effect and the hydrotropic effect of MCHM would be the primary effect on the transcriptome and metabolome, especially at early time points.

Among the differentially expressed pathways in the *med15* mutant, *PHO89* was noted because Pho89 is a high-affinity transporter that is induced in inorganic-phosphate-limiting conditions [74] and intracellular levels of phosphate increase in MCHM exposure [27]. The other high-affinity inorganic phosphate transporter is Pho84, which is also induced in phosphate-limiting conditions [75]. *PHO84* expression was down-regulated fourfold in the *med15* mutant grown in YPD but was not significantly different in other strains and conditions. While *PHO89* and *PHO84* are both induced in phosphate-limiting conditions, the kinetics are slightly different due to the different transcription factors and kinases that regulate their expression [76]. Pho2 and Pho4 are transcription factors that regulate the PHO regulon and the expression of secreted acid phosphatases; Pho5, Pho11, and Pho12 were also increased in the MCHM treatment. Furthermore, there are other signaling pathways that affect the regulation of *PHO89,* such as Snf1, which phosphorylates Mig1 and Nrg1 under stress and regulates *PHO89* expression but not *PHO84* expression [76]. Pho84 and Pho89 have nonredundant roles in the MCHM response. Despite *PHO89* being differentially expressed, only the *pho84* mutant was sensitive to MCHM. It is likely that Med15 directly regulated *PHO84* expression because it physically binds the *PHO84* promoter, and it is not found at the *PHO89* promoter [51].

The multiple bands of Med15 proteins shift between the different alleles. To visualize Med15, the 13xMyc tag was integrated at the C-terminal end. While the multiple bands may represent N-terminal degradation products, most of these bands migrate slower than the predicted size of full-length Med15-Myc. Several lines of evidence pointed us to investigate whether Snf1 regulates Med15. Snf1 has been copurified with the Mediator complex [36]. Med15 has several phosphorylations in the C-terminal MAD. The *snf1* mutant is MCHM sensitive [57] and Snf1 has a role in regulating *PHO89* [76]. In the *snf1* knockout, the impact of changes in the polyQ tract was only seen when Med15 was Myc tagged in MCHM treatment. In calcofluor white, caffeine, DTT, and hygromycin treatments, the loss of Snf1 flipped the response of strains carrying untagged alleles of Med15. Med15^S288c^ grew better than yeast with Med15^YJM789^ in these stresses, while the yeast with Med15^YJM789^ and *snf1* was more resistant in hydrogen peroxide. In Western blots, the pattern of Med15 bands did not change in *snf1* knockout but it did change in the *reg1* knockout. Snf1 can be overactivated by knocking out Reg1, the repressor of SNF1. We did not address the nature of the different bands but they are different from N-terminally tagged HA-Med15 [10], which looked more like the band patterns of Med15^YJM789^ but ran true to size compared to Med15^YJM789^. Other studies have used Med15^BY^-Myc for Western blots but the studies cropped the Western blots, therefore the pattern of bands could not be compared. However, Med15-Myc decreased the association of the rest of the Mediator tail subunits [77]. This leads to the hypothesis that a Myc tag on any component of the tail weakens the interaction with the rest of the complex [47]. However, a decreased association of Med15-Myc with the Mediator complex does not explain all the results presented here. Yeast with Med15^YJM789^-Myc in BY4741 and hemizygous knockouts were more sensitive than *med15* knockouts to MCHM. If the stoichiometry of the Mediator was the only contributing factor to differences, then the phenotypes of the knockouts should show a more extreme version of hypomorphic alleles. This was directly tested via direct comparisons of growth between strains with different Med15 alleles with and without Myc tagged in diverse chemicals. The Myc tag can serve as an in vitro target of SNF1 [35], and therefore if SNF1 is already associated with the Mediator, then the Myc tag may increase the SNF1-dependent phosphorylation of the Mediator.

The differences between Med15 alleles could only be seen in MCHM when Med15 was tagged with Myc, while the impact of the tag sometimes exaggerated or lessened differences in other stresses, and sometimes had no impact. The original Myc tag was derived from a peptide from human c-Myc [78]. Myc is a family of oncogenic transcription factors. Of the multiple peptides tested, only 9E10 did not cross-react with the c-myc from other organisms and had a low background on Western blots [78]. The 13xMyc epitope tag used here was tandem repeats of EQKLISEEDL [79]. Tagging a protein can affect the folding, localization, and association with other proteins in a complex. Under normal conditions, Med2, Med3, and Med15 can be recruited to chromatin, independent of the rest of the Mediator complex [77,80,81]. From the recent structures of the Mediator, Med2 and Med3 bound the C-terminal tail of Med14 in the middle and directly bound Med15. Med15, in turn, bound Med16, and Med5 is at the very distal end of the tail (Figure 1A, [3]). The Med15–Med5–Med16 complex is posited to have a function independent of the full Mediator complex [48]. The Myc tag does not include the basic helix turn domain common in transcription factors that binds DNA. Other TFs regulated by the mediator, such as Pho4, share homology with the DNA-binding domains of Myc proteins [9].

## 4. Materials and Methods

### 4.1. Strain Construction and Cloning

All strains and sources are in Appendix A. Med15 sequences were extracted from the resequenced genomes of BY4741, BY4742, AWRI1631, RM11-1a, and YJM789 [82]. Med15 was tagged at the C-terminus with a 13xMyc tag, with KanR as the selectable marker in S288c (GSY147), BY4741, and YJM789, as previously described [37,79]. *MED15* has a polymorphism just after the stop codon, therefore allele-specific primers were used for the 3′ Myc tagging and knocking out (Appendix A). Primers to the genomic *MED15* amplified 499 nucleotides upstream from the start and a 3′ tagging primer was used to include the promoter, coding region, Myc tag, and KanR marker. The PCR product was then cloned into the NotI restriction site in pRS316. *MED15* was knocked out in YJM789K5a (isogenic with YJM789, except as a MATa prototroph), and then backcrossed to generate YJM789K6alpha, as previously described [83]. The KanR marker of BY4741 knockout yeast of *snf1, reg1,* and *ydj1* [84] was switched with HygR, crossed with BY4742 *med15::NatR* to generate double mutants, and then transformed with plasmids containing different alleles of *MED15.*

Because of the lack of convenient restriction sites and the repetitive nature, *MED15* domain swaps of the alleles proved challenging. Med15 domain swaps were carried out using PCR and then gap repair transformation of plasmid encoding *MED15^YJM789^-Myc* was used with polyQ inserts from *MED15^S288c^.* We used inverted PCR to amplify the *pMED15^YJM789^-Myc* plasmid to linearize the vector with gaps at each of the polyQ repeats. Each polyQ tract from *MED15^S288c^* was independently amplified with between 20 and 180 nucleotides of homology with the PCR-amplified vector. The vectors lacking polyQI or polyQII from the *pMED15^YJM789^-Myc* plasmid were amplified using primers that amplified around the plasmid leaving a gap at the polyQI or polyQII site. The linearized plasmid from the PCR amplifiation had between 20 and 180 nucleotides of overlap at the 5′ and 3′ ends around the region containingthe polyI or polyQII. The PCR product containing polyQI or polyQII was transformed with the PCR-amplified vector lacking polyQI or polyQII. Cloning was carried out via a gap repair transformation [85]. The inserts were amplified separately with flanking homology to the region around the vector’s 5′ and 3′ ends. The insert and linear vectors were transformed into BY4741 *med15* yeast, and transformants were selected on YPD with G418 based on the colony size, as *med15* yeast are slow-growing, in combination with selecting for markers on the plasmid. Genomic DNA was extracted and transformed into DH10 beta *Escherichia coli* from NEB and then retransformed into BY4741 *med15.* All plasmids were verified using Sanger sequencing. Plasmids were rescued via passaging through *E. coli* and inserts were verified. All primers are listed in Appendix A.

### 4.2. Growth Conditions

Plasmids were maintained with the addition of 0.5 mg/mL G418 from Invivogen in YPD. In minimal media (YM), plasmids were maintained by supplementing the media with uracil, histidine, and methionine, or by switching the nitrogen source to glutamate (MSG), and then adding G418 with amino acids as needed. All amino acids were purchased from Sigma. Yeast were grown in liquid media as indicated to mid-log phase, 550 ppm MCHM was added to YPD (650 ppm was added to YM), and cells were harvested after 30 minutes of exposure. Western blots were carried out as previously described [37]. Solid media plates were cooled to 65 °C before MCHM was added and gently mixed until dissolved. Plates were used within 24 h to limit the evaporation of MCHM. The yeast was serially diluted 10-fold and spotted onto solid media. Plates were photographed after 2–3 days of growth. For multiple drug screening in the TECAN, the automated plate reader, yeast were grown to stationary phase and then diluted to 0.1 OD with appropriate drugs and read at OD_600_ [83]. The following chemicals were added: 3 mM hydrogen peroxide (H_2_O_2_), 0.25 g/mL 4-nitroquinoline 1-oxide (4NQO), 400 ppm 4-methylcyclohexanol (MCHM), 1 mM copper sulfate (CuSO_4_), 7.5 ng/mL rapamycin (Rapa), 0.1% glyphosate (CR41), 100 mM hydroxyurea (HU), 20 µg/mL camptothecin (CPT), 8.5 mM beta-mercaptoethanol (βME), 5 mM calcofluor white (CALC), 2.5 mM caffeine (CAFF), 20 mM dithiothreitol (DTT), and 50 µg/mL hygromycin. Cells were grown with readings taken every hour. During log-phase, the OD_600_ of yeast carrying *MED15^S288c^* was subtracted from *MED15^YJM789^* at the point of maximal growth difference.

### 4.3. Transcriptomics

RNAseq was carried out in biological triplicate from yeast grown in YM supplemented with histidine, leucine, and methionine or YPD with G418. PolyA RNA was selected using a Karpa Stranded RNAseq library preparation kit according to the manufacturer’s instructions (catalog number KK8401). Libraries were sequenced on an Illumina PE50bp high output flowcell. Basecalls were performed with Illumina’s FASTQ Generation (v1.0.0) available in BaseSpace. Transcripts quantification was done with salmon (v0.9.1) versus the transcripts file BY4741_Toronto_2012_cds.fsa (available from https://downloads.yeastgenome.org/sequence/strains/BY4741/BY4741_Toronto_2012/). This data is available from GSE, accession number GSE129898 (https://www.ncbi.nlm.nih.gov/geo/query/acc.cgi?acc=GSE129898). Quantification tables were imported to R (3.4.4) and gene-level analysis was created with the tximport (1.6.0) package. For the transcripts to gene translation the homemade R package TxDb.Scerevisiae.SGD.BY4741 was used. This package was built from the BY4741_Toronto_2012.gff file using GenomicFeatures (1.30.3). The gene differential expression analysis and the data quality assessment were done with DESeq2 (1.18.1). The *p*-values were adjusted to an false discovery rate (FDR) of 0.005. The microarray (MA)-plots were done with ggpubr (0.1.6).

GO term analysis was carried out with clusterProfiler [86] (3.6.0). The open reading frames (ORF) names from genes that were up- or down-regulated in each condition were translated to the corresponding Entrez id using the function bitr and the package org.Sc.sgd.db. The resulting gene clusters were processed with the compareCluster function, in enrichGO mode, using org.Sc.sgd.db as a database, with Biological Process ontology, with cutoffs of *p*-value = 0.01 and *q*-value = 0.05, adjusted by FDR, to generate the corresponding GO profiles, which were then simplified with the function simplify. The simplified profiles were represented as dotplots showing up to 15 more relevant categories.

### 4.4. Western Blot

Proteins were extracted, immunoprecipitated, separated in 5%–12% SDS-PAGE, and transferred onto 0.2-micron PVDF, as previously described [37]. Antibodies were diluted into freshly made 3% BSA Fraction V in TBS-Tween. ECL kit and HRP secondary antibodies were used to visualize mouse anti-Myc E910 (1:7500) from various manufacturers and rabbit anti-PGK (1:10,000) on a Protein Simple using the default chemiluminescence setting.

### 4.5. Flow Cytometry

BY4741 cells were grown to saturation overnight and returned to mid-log phase. Cells were then diluted to a starting OD_600_ of 0.3 in biological triplicate in YPD media containing MCHM. For the measurement of ROS, live cells were pelleted, then suspended in 200 μL of 50 mM DHE in phosphate-buffered saline (PBS). The dyed cultures were incubated at 30 °C for 20 min and washed with PBS. A positive control sample of BY4741 cells was treated with 25 mM H_2_O_2_ for 1.5 h. The DHE-dyed samples were then analyzed within 2 h of harvesting on a BD LSRFortessa using preset propidium iodide detection defaults. Approximately 30,000 events were collected per sample for downstream analysis.

## 5. Conclusions

By their nature, the structure of intrinsically disordered regions are difficult to determine and are important for changes in protein complex conformations [22,28,87,88,89]. The fuzzy/IDR domains of Med15 and the expansions of the polyQ tracts increased phenotypic diversity. Rim101, a transcription factor with a polyQ tract, affects allele-specific expression in one strain background but not the others tested [90]. There are multiple phosphorylations in Med15 that regulate the transcriptional response to stress [6]. Expression of the longer Med15 allele changed the response to MCHM as other polymorphic transcription factors change the response to other chemical stressors [37]. Variation in key regulators permits the expression of cryptic genetic variation to alter phenotypes. These proteins are master variators.

## Figures and Tables

**Figure 1 ijms-21-01894-f001:**
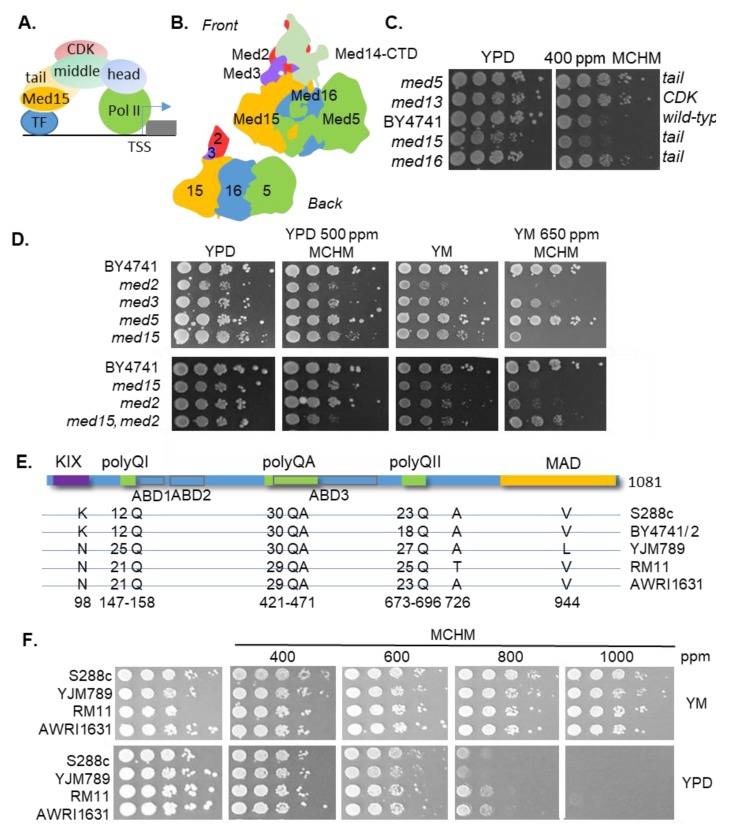
Role of the Mediator tail in response to 4-methylcyclohexane methanol (MCHM). (**A**) Schematic of the Mediator complex. Med15 as part of the tail subcomplex directly interacts with transcription factors (TFs). The middle of the Mediator complex tethers the CDK (cyclin-dependent kinase). The head directly interacts with RNA polymerase II (Pol II) at promoter regions to initiate transcription at transcriptional start sites (TSSs) of genes (gray box). (**B**) Representation of protein components of the Mediator tail based on structures and modeling [3]. Med2 (red), Med14-CTD (light green), Med3 (purple), Med15 (orange), Med16 (blue), and Med5 (green) comprise the tail of the Mediator complex. From the back view, Med5, Med16, Med15, Med3, and Med2 (in order from farthest to closest to the rest of the Mediator complex) are associated with Med14 (not pictured here). (**C**) Growth assays of yeast with different components of the Mediator knocked out in BY4741 grown with and without 400 ppm MCHM in rich media (YPD). (**D**) Growth assays of yeast with different components of the Mediator tail knocked out in BY4741 grown with and without 500 ppm MCHM or 650 ppm in YPD or minimal media supplemented with lysine (YM) + histidine, uracil, leucine and methionine (HULM), respectively. (**E**) Schematic of the Med15 protein. Above the blue line are polymorphic domains including the KIX domain, polyglutamine domain I (polyQI), polyglutamine/alanine domain (polyQA), polyglutamine domain II (polyQII), and the Mediator activation/association domain (MAD). Under the blue line are the fuzzy domains represented as ABD1-3 (activator-binding domains) in gray outlined boxes [10,11]. The Med15 polymorphic amino acids are drawn below from five genetically diverse yeast. Amino acid numbers are based on S288c. (**F**) Growth assays of genetically diverse yeast strains in the presence of MCHM on different growth media with increasing concentrations of MCHM. Yeast were spotted in ten-fold dilutions onto YM or YPD. RM11, S288c (GSY147), and AWRI1631 are MATa prototrophs while YJM789 is a MATalpha *lys2* strain.

**Figure 2 ijms-21-01894-f002:**
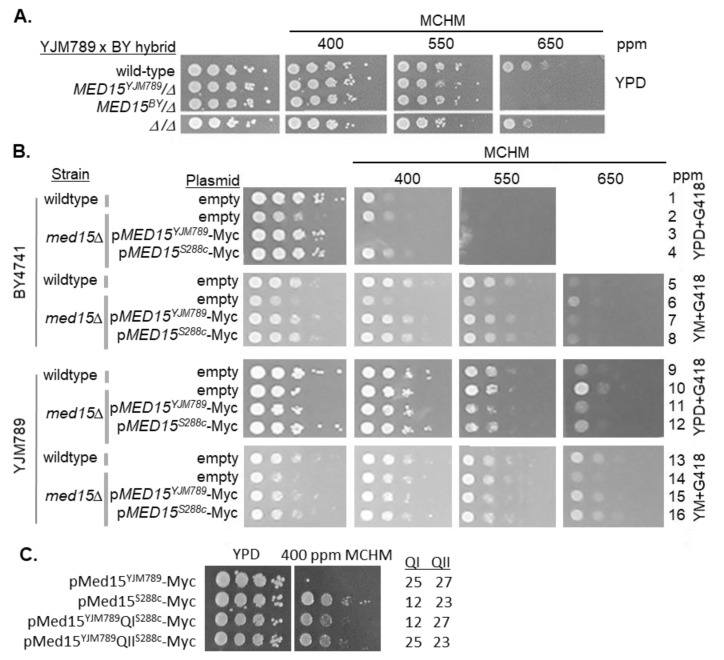
Genetic variation in Med15 contributed to variation in the MCHM response. (**A**) Reciprocal hemizygotes of Med15 in BY4741×YJM789 hybrids were grown on MCHM in YPD. Med15 was tagged at the chromosomal locus with 13xMyc at the C-terminal end or knockout with a dominant drug marker in haploid parents. The yeast was then mated and diploids were selected. (**B**) Med15 allele swap in BY4741 and YJM789 was carried out by cloning Med15-13xMyc with *KanR* onto pRS316. Med15 plasmids were transformed into the wildtype and *med15::NatR* stains in the BY4741 and YJM789 (YJM789K5a, a MATa prototroph) backgrounds. Plasmids were maintained via growth on YPD with G418. Glutamate was used as the nitrogen source in minimal media (YM) with histidine, uracil, leucine, and methionine to supplement BY4741 such that G418 would be selective and maintain the plasmid. The empty plasmid was pGS35 (*KanR*). (**C**) Growth of BY4741 *med15* mutants expressing polyQI and polyQII domain swaps in Med15^YJM789^-Myc. The length of each polyQ for each allele is noted to the right of the figure.

**Figure 3 ijms-21-01894-f003:**
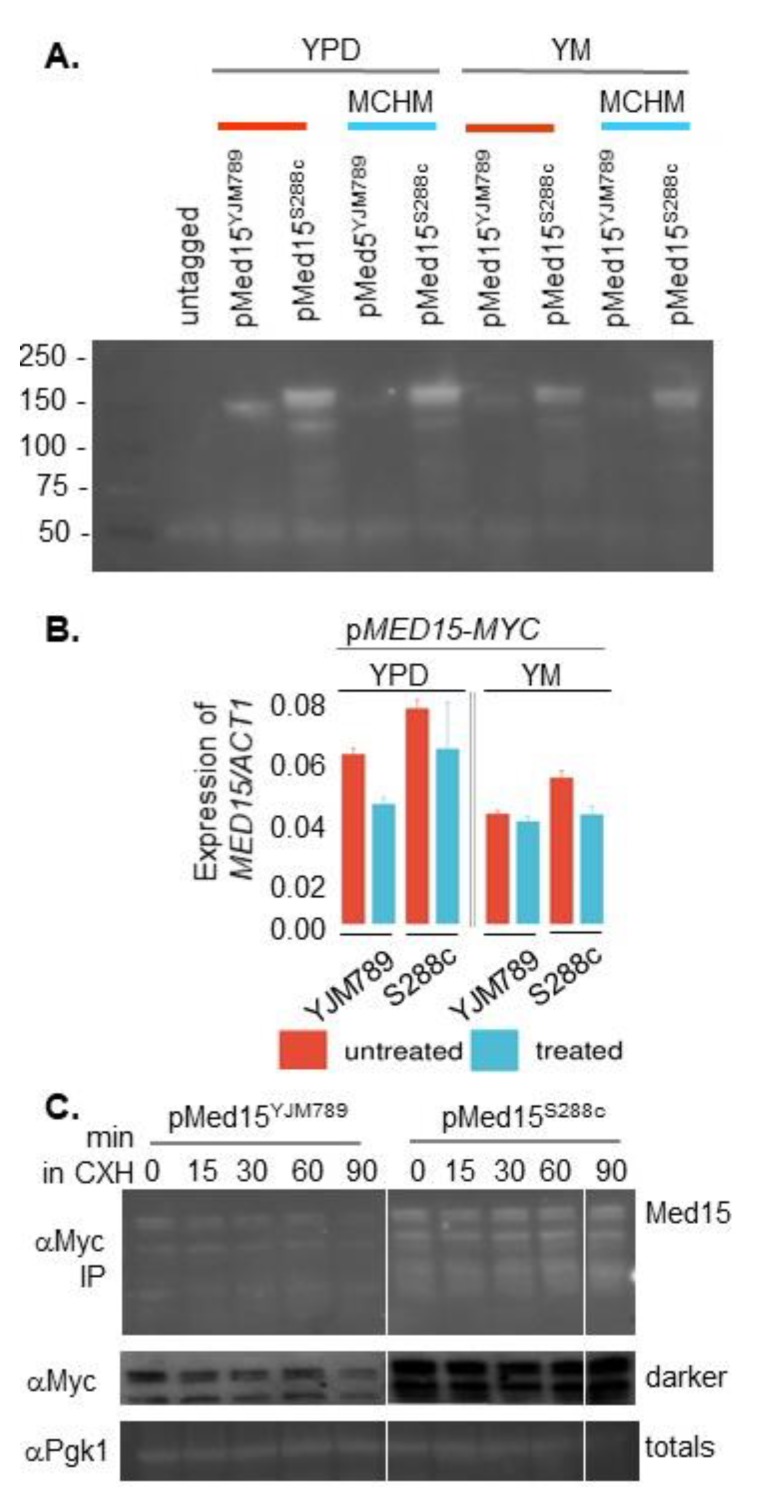
Changes in the expression levels of different alleles of Med15 treated with MCHM. (**A**) Protein levels of Med15-13xMyc expressed from a plasmid in BY4741 *med15* yeast. The yeast was grown in selective media until mid-log and then shifted to 550 ppm MCHM for 90 minutes. Med15-13xMyc was immunoprecipitated from equal amounts of protein extract. (**B**) mRNA levels of *MED15* expressed from a plasmid in BY4741 *med15* yeast normalized to *ACT1* mRNA. Transcript levels were extracted from RNAseq data. The yeast was grown in YPD (with G418) or YM (yeast minimal media supplemented with histidine, leucine and methionine (HLM) and then treated with 550 ppm MCHM for 30 minutes. (**C**) Western blot of Med15-Myc immunoprecipitated from BY4741 carrying YJM789 and S288c alleles of Med15 on single copy plasmids from yeast grown in YPD at 0, 15, 30, and 90 minutes after the addition of cycloheximide. The total lysate was run separately and Pgk1 was blotted as a loading control.

**Figure 4 ijms-21-01894-f004:**
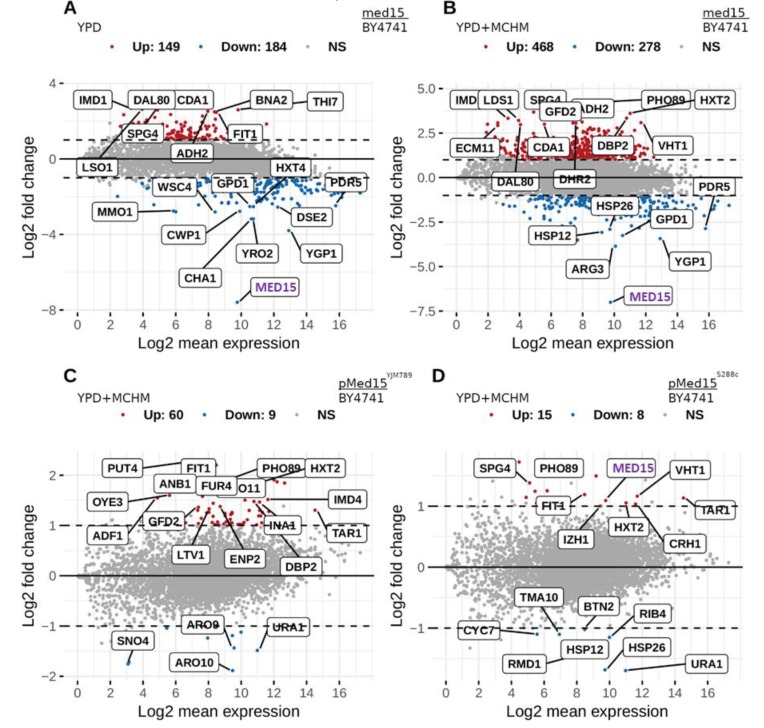
Changes in the transcriptome of BY4741 yeast carrying different alleles of Med15 treated with MCHM and grown in YPD. (**A**) Differentially expressed mRNA from wildtype yeast (BY4741) compared to a *med15* knockout strain grown in YPD. (**B**) Differentially expressed mRNA from wildtype yeast (BY4741) compared to a *med15* knockout strain grown in YPD, then shifted to 400 ppm MCHM for 30 minutes displayed on a log scale. (**C**) Differentially expressed mRNA from wildtype yeast (BY4741) compared to a *med15* knockout strain carrying Med15^YJM789^ expressed from a plasmid grown in YPD, then shifted to 550 ppm MCHM for 30 minutes. (**D**) Differentially expressed mRNA from wildtype yeast (BY4741) compared to a *med15* knockout strain carrying Med15^S288c^ expressed from a plasmid grown in YPD with G418, then shifted to 550 ppm MCHM for 30 min.

**Figure 5 ijms-21-01894-f005:**
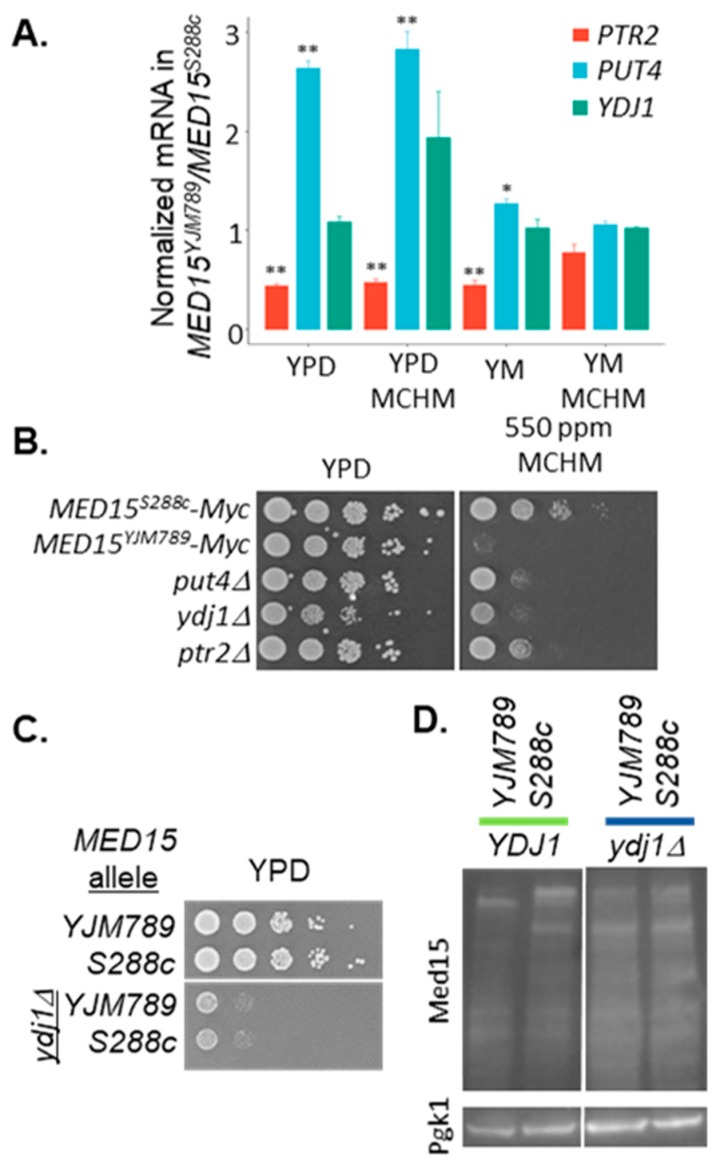
Conditions that affected the stability of Med15. (**A**) Expression levels of *PTR2*, *PUT4*, and *YDJ1* extracted from RNAseq data from Appendix A. (**B**) Plasmids containing Med15^YJM789^-Myc and Med15^S288c^-Myc were transformed into single mutants of *med15.* The *put4*, *ptr2*, and *ydj1* in the BY4741 background were grown and serial dilutions of yeast on YPD were grown for 2 days at 30 °C and then photographed. (**C**) Serial dilution of yeast knockouts of *ydj1* yeast expressing YJM789 or S288c alleles of Med15-Myc on YPD. (**D**) Western blot of YJM789 or S288c alleles of Med15-Myc immunoprecipitated from BY4741 or the *ydj1* mutant, which were grown in YPD.

**Figure 6 ijms-21-01894-f006:**
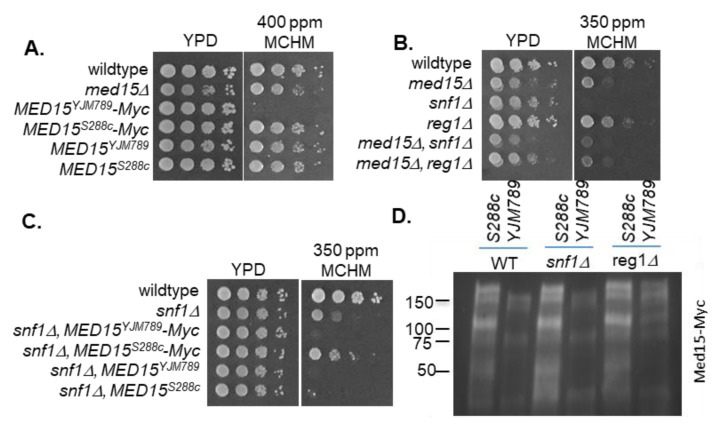
Impact of Snf1 and Reg1 on yeast expressing different Med15 alleles. (**A**) Serial dilutions of BY4741 with different alleles of Med15 that were untagged or C-terminally tagged with 13xMyc grown on YPD or 400 ppm of MCHM and photographed after three days of growth. (**B**) Serial dilution of BY4741 with single and double mutants containing *med15Δ*, *snf1Δ*, and *reg1Δ* knockouts grown on YPD or 350 ppm of MCHM. Plates were photographed after two days of growth. (**C**) Serial dilutions of BY4741 *snf1Δ* mutants expressing different alleles of Med15 with and without the 13xMyc tag grown on YPD or 350 ppm of MCHM. Plates were photographed after three days of growth. (**D**) Western blot of YJM789 or S288c alleles of Med15-Myc immunoprecipitated from BY4741 grown in YPD with *SNF1* or *REG1* deleted.

**Figure 7 ijms-21-01894-f007:**
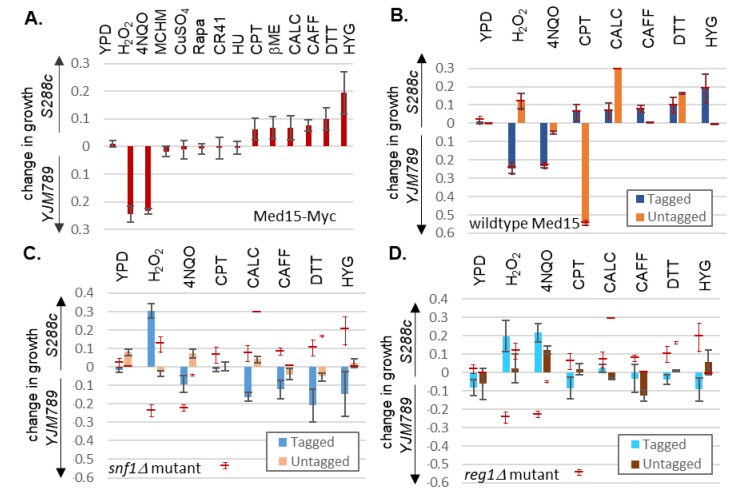
Quantitative growth assays of BY4741 med15 carrying different alleles of Med15 in snf1 and reg1 mutants with different drugs. At the time, when there was the maximum growth difference, the OD_600_ of yeast carrying Med15^S288c^ was subtracted from the OD_600_ of yeast carrying Med15^YJM789^. Values above the y-axis indicate increased growth of the yeast with Med15^S882c^ and values below the y-axis indicate that the yeast with Med15^YJM789^ grew better than yeast with the other allele. The following chemicals were added: hydrogen peroxide (H_2_O_2_), 4-nitroquinoline 1-oxide (4NQO), 4-methylcyclohexanol (MCHM), copper sulfate (CuSO_4_), rapamycin (Rapa), glyphosate (CR41), hydroxyurea (HU), beta-mercaptoethanol (βME), calcofluor white (CALC), caffeine (CAFF), dithiothreitol (DTT), and hygromycin (HYG). (**A**) Growth of BY4741 med15 yeast carrying different alleles of Med15-Myc. (**B**) Growth of BY4741 med15 yeast carrying different alleles of Med15 (untagged, blue) or Med15-Myc (tagged, orange). (**C**) Growth of BY4741 med15 and snf1 yeast carrying different alleles of Med15 (untagged, light blue) or Med15-13xMyc (tagged, light orange). The respective values of wildtype yeast from panel (B) are shown as a thin red line with the standard deviation range shown as thin red lines. (**D**) Growth of BY4741 med15 and reg1 yeast carrying different alleles of Med15 (untagged, cyan) or Med15-Myc (tagged, brown). The respective values of wildtype yeast from panel (B) are shown as a thin red line with the standard deviation range shown as thin red lines.

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
