# Peer review of "The Polymorphic PolyQ Tail Protein of the Mediator Complex, Med15, Regulates the Variable Response to Diverse Stresses"

_ijms, 2020, doi:10.3390/ijms21051894_

Round 1

Reviewer 1 Report

In this manuscript, the authors attempt to study the effect of the Mediator Complex, Med15 to variable response to diverse stresses.   This study used yeast as a model but did not show how would it apply to human. It would be helpful to show how stress response would affect Med15 or its homolog in human cells. Does changes in Med15 (by phosphorylation) affect protein aggregation and if that caused changes in autophagy

Author Response

R1.Comment1 In this manuscript, the authors attempt to study the effect of the Mediator Complex, Med15 to variable response to diverse stresses. This study used yeast as a model but did not show how would it apply to human. It would be helpful to show how stress response would affect Med15 or its homolog in human cells.

R1.Answer1 We appreciate the reviewer’s comments and will further elaborate on human applications. However, in the five days for the resubmission we don’t have time to carry out more experiments exploring the role of the S. cerevisiae or the human homolog of Med15 in human cells. We did expand on the similarities and differences between the two orthologs. “Human Med15 is 19.9% glutamines compared to the yeast ortholog that has 16.6% (Supplemental Figure S1). The yeast Med15S288c 27% longer. While 47% of human orthologs tested can complement yeast knockouts, the percentage decreases for transcription factors and other proteins that associate with DNA (Kachroo et al. 2015).” We have added text in the manuscript line XX and new supplemental figure S1. Line 142-148

R1.C2 Does changes in Med15 (by phosphorylation) affect protein aggregation…

R1.A2 We have been currently addressing if the multiple bands of Med15 seen in the gels are from phosphorylations but after phosphatase treatment did not shift the bands. We are currently purifying Med15 to map post-translational modifications, but we have not yet run the samples. Med15 is a large protein, with a predicted size of 120 kDa. There are two studies that addressed the aggregation. One could only detect aggregation after overexpression for 48 hours (Alberti et al. 2009) and another only expressed the first Q1 tract, truncating most of the protein. The large C-terminal domain could induce the aggregation prone N-terminal portion containing the two polyQ tracts and the fuzzy domains (ABD) to remain in solution. Therefore, using known techniques that can detect aggregation, the cells would have to be manipulated in such away would likely alter the function of Med15 and the cellular response to MCHM and the other stresses that we explored. Full length Med15 forms foci in response to rapamycin but whether those proteins are aggregated are not was not determined (Zhu et al. 2015). We note that there are several known phosphorylations, but the functional significance as not been determined.

R1.C3 ...and if that caused changes in autophagy.

We have not investigated the role of autophagy in MCHM response or if Med15 has a role in regulating this process. We are continuing our research on MCHM and will likely assess if there is a role for autophagy in the future.

Alberti, S., R. Halfmann, O. King, A. Kapila, and S. Lindquist, 2009 A systematic survey identifies prions and illuminates sequence features of prionogenic proteins. Cell 137: 146–158.

Ansari, S. A., M. Ganapathi, J. J. Benschop, F. C. P. Holstege, J. T. Wade et al., 2011 Distinct role of Mediator tail module in regulation of SAGA‐dependent, TATA‐containing genes in yeast. EMBO J. 31: 44–57.

Grünberg, S., S. Henikoff, S. Hahn, and G. E. Zentner, 2016 Mediator binding to <scp>UAS</scp> s is broadly uncoupled from transcription and cooperative with <scp>TFIID</scp> recruitment to promoters. EMBO J. 35: 2435–2446.

Jeronimo, C., and F. Robert, 2014 Kin28 regulates the transient association of Mediator with core promoters. Nat. Struct. Mol. Biol. 21: 449–455.

Jeronimo, C., S. Watanabe, C. D. Kaplan, C. L. Peterson, and F. Robert The Histone Chaperones FACT and Spt6 Restrict H2A.Z from Intragenic Locations.:

Kachroo, A. H., J. M. Laurent, C. M. Yellman, A. G. Meyer, C. O. Wilke et al., 2015 Systematic humanization of yeast genes reveals conserved functions and genetic modularity. Science (80-. ). 348: 921–925.

Zhu, X., L. Chen, J. O. P. Carlsten, Q. Liu, J. Yang et al., 2015 Mediator tail subunits can form amyloid-like aggregates in vivo and affect stress response in yeast. Nucleic Acids Res. 43: 7306–14.

Reviewer 2 Report

Mediator is a transcriptional coactivator complex that relays regulatory information from transcription factors bound to distal regulatory elements to the promoter-bound RNA polymerase II pre-initiation complex (PIC). The interaction of the Mediator tail module with activators is particularly important for budding yeast exposed to various stressors. The tail subunit Med15 is a primary site of activator interaction with Mediator, and contains two polyQ tracts that participate in interactions with Gcn4, induced in response to amino acid starvation. Here, Gallagher et al report a characterization of the effects of polyQ tract length polymorphisms on stress responsiveness and Med15 protein levels. It’s a very interesting study and the results seem to speak for themselves; I think it will make a nice addition to the literature on Med15. However, I had a hard time reading the paper itself due to organization issues, outlined below.

Comments:

The introduction is very detailed, and while this isn’t inherently a bad thing, I found it somewhat hard to follow and repetitive in places. For instance, the interaction of Med15 with Gcn4 via fuzzy domains is described both on lines 65-68 and 87-88. The transition to the description of Ydj1 was also a bit jarring, and the introduction of hydrotropes and MCHM was very long. The description of how Mediator works is sprinkled throughout the introduction and seems scattered. I think a condensation and refocusing of the introduction is needed to enhance its readability and better present the existing literature, rationale for the study, and its primary results. The organizational issues continue in the results, with large sections of background information on Med15 structure repeated from the introduction (for instance, lines 165-173) between results descriptions.

Line 34: some clarification of the terminology for discussing Mediator structure may be warranted here. Mediator’s subunits are generally said to be classified into ‘modules’ versus domains, and the ‘body’ is generally referred to as the ‘middle.’

In Fig. 1C, the authors show the growth of WT BY4741 and Mediator KO (med5, 13, 15, and 16) yeast on MCHM. The results appear to indicate that loss of Med15 slightly impairs growth with MCHM while removal of 5, 13, and 16 enhances growth. While the use of med13 is interesting because of the possible role of the kinase module in antagonizing the tail, I think the authors should also test at least one of the two remaining tail subunits (Med2/3), as Med5 and Med16 do not necessarily behave as standard tail subunits (deletion of Med16 removes a stable triad of Med2/3/15 from core Mediator, while Med5 seems to have anti-activation properties).

It is repeatedly stated that Myc tagging has been shown to affect Med15 function; some citations would be good (e.g. PMID 21971086, 15254252).

It is stated on lines 183-184 that YJM780 and BY4741 were selected for further analysis because they represent the extremes of MCHM resistance. Where was the sensitivity of these strains directly compared? It seems that BY4741 was excluded from Fig. 1E.

Since the Mediator tail module has been reported to preferentially SAGA-dominated genes, did the authors attempt to classify their differentially expressed genes using the SAGA and TFIID-dominated categories?

Line 336: unpublished data is cited for genome-wide localization of Med15. There have been many published studies on the genomic binding of Mediator (e.g. PMID 24704787, 25959393, 27797823).

Author Response

R2.C1 The introduction is very detailed, and while this isn’t inherently a bad thing, I found it somewhat hard to follow and repetitive in places. For instance, the interaction of Med15 with Gcn4 via fuzzy domains is described both on lines 65-68 and 87-88. The transition to the description of Ydj1 was also a bit jarring, and the introduction of hydrotropes and MCHM was very long. The description of how Mediator works is sprinkled throughout the introduction and seems scattered. I think a condensation and refocusing of the introduction is needed to enhance its readability and better present the existing literature, rationale for the study, and its primary results. The organizational issues continue in the results, with large sections of background information on Med15 structure repeated from the introduction (for instance, lines 165-173) between results descriptions.

R2.A1 The Introduction has been condensed. I deleted the repetitive sections and reorganized the introduction. Line 46-57, 59-68, 92-94

R2.C2 Line 34: some clarification of the terminology for discussing Mediator structure may be warranted here. Mediator’s subunits are generally said to be classified into ‘modules’ versus domains, and the ‘body’ is generally referred to as the ‘middle.’

R2.A2 The document and all the figures have been edited so that terminology is consistent with the field.

R2.C3 In Fig. 1C, the authors show the growth of WT BY4741 and Mediator KO (med5, 13, 15, and 16) yeast on MCHM. The results appear to indicate that loss of Med15 slightly impairs growth with MCHM while removal of 5, 13, and 16 enhances growth. While the use of med13 is interesting because of the possible role of the kinase module in antagonizing the tail, I think the authors should also test at least one of the two remaining tail subunits (Med2/3), as Med5 and Med16 do not necessarily behave as standard tail subunits (deletion of Med16 removes a stable triad of Med2/3/15 from core Mediator, while Med5 seems to have anti-activation properties).

R2.A3 We have added a figure (Figure 1D) which includes the med2, med3, med5, and med15 mutants. Although, the dose of MCHM is slightly higher, and the cells were grown a day longer to compensate for the higher dose. Mutants in the other tail subunits, med2 and med3 are also MCHM sensitive in YPD. In YM, the order of most sensitive to least sensitive is med2, med3, med15, and med5. We also constructed a med2, med15 double mutant which was slightly more sensitive on YPD and paradoxically was more resistant in YM to MCHM. Line142-148

R2.C4 It is repeatedly stated that Myc tagging has been shown to affect Med15 function; some citations would be good (e.g. PMID 21971086, 15254252).

R2.A4 The study noted Myc-tagged Med15 but didn’t show a western blot and did only ChIPped Med15 in unpertubed cells (Ansari et al. 2011). They did not note any differences in in tagged versus untagged. The data was only available in the raw format from another study (Jeronimo and Robert 2014) and we didn’t not have sufficient time to remap and reanalyze their data.

R2.C5 It is stated on lines 183-184 that YJM780 and BY4741 were selected for further analysis because they represent the extremes of MCHM resistance. Where was the sensitivity of these strains directly compared? It seems that BY4741 was excluded from Fig. 1E.

R2.A5 BY4741 contains auxotrophies that prevent growth in unsupplemented YM. We have added a supplemental figure (Figure S1A) with BY4741, S288c, and YJM789 grown on the same plates for direct comparison. Line 170-171

R2.C6 Since the Mediator tail module has been reported to preferentially SAGA-dominated genes, did the authors attempt to classify their differentially expressed genes using the SAGA and TFIID-dominated categories?

R2.A6 Suggested papers, particularly "Huisinga, Kathryn L, and B Franklin Pugh. 2004. “A Genome-Wide Housekeeping Role for TFIID and a Highly Regulated Stress-Related Role for SAGA in Saccharomyces Cerevisiae.” Molecular Cell 13 (4): 573–85. https://doi.org/10.1016/s1097-2765(04)00087-5.", they provided a supplementary table where all genes are labelled depending on their SAGA/TFIID dominated status. We used this table to look for the appropriated annotation for the 1111 unique genes that are differentially expressed in any of the comparisons we made for the paper (RNA-Seq data). From these, 1012 genes can be found in their supplementary table. We have added a pie chart with the labelling of our relevant genes (Figure S8). As can be seen the SAGA-dominated set is a minority of a 26%, but given that the paper a cited above claims that SAGA-dominated genes are just 10% of the genome, then we do have an enrichment of those genes (of 2.6X compared to a randomly sampled set of genes of equivalent size). Line 302-311

R2.C7 Line 336: unpublished data is cited for genome-wide localization of Med15. There have been many published studies on the genomic binding of Mediator (e.g. PMID 24704787, 25959393, 27797823).

R2.A7 We used our own unpublished ChIP-seq data to select candidate Med15 targets. The study that ChIPped Med145-9Myc tag did not note any changes in stress response or protein isoforms by western blot (Jeronimo and Robert 2014) nor did they publish the list of associated genes but only a heatmap (Jeronimo and Robert 2014). In the next study they addressed the impact of histone chaperones on the genomic localization of H2A.Z and does not address the Mediator localization (Jeronimo et al.). In the last suggested study (Grünberg et al. 2016), the reanalyzed from the first suggested publication (Jeronimo and Robert 2014). The data was only available in the raw format and we didn’t not have sufficient time to remap and reanalyze their data.

Alberti, S., R. Halfmann, O. King, A. Kapila, and S. Lindquist, 2009 A systematic survey identifies prions and illuminates sequence features of prionogenic proteins. Cell 137: 146–158.

Ansari, S. A., M. Ganapathi, J. J. Benschop, F. C. P. Holstege, J. T. Wade et al., 2011 Distinct role of Mediator tail module in regulation of SAGA‐dependent, TATA‐containing genes in yeast. EMBO J. 31: 44–57.

Grünberg, S., S. Henikoff, S. Hahn, and G. E. Zentner, 2016 Mediator binding to <scp>UAS</scp> s is broadly uncoupled from transcription and cooperative with <scp>TFIID</scp> recruitment to promoters. EMBO J. 35: 2435–2446.

Jeronimo, C., and F. Robert, 2014 Kin28 regulates the transient association of Mediator with core promoters. Nat. Struct. Mol. Biol. 21: 449–455.

Jeronimo, C., S. Watanabe, C. D. Kaplan, C. L. Peterson, and F. Robert The Histone Chaperones FACT and Spt6 Restrict H2A.Z from Intragenic Locations.:

Kachroo, A. H., J. M. Laurent, C. M. Yellman, A. G. Meyer, C. O. Wilke et al., 2015 Systematic humanization of yeast genes reveals conserved functions and genetic modularity. Science (80-. ). 348: 921–925.

Zhu, X., L. Chen, J. O. P. Carlsten, Q. Liu, J. Yang et al., 2015 Mediator tail subunits can form amyloid-like aggregates in vivo and affect stress response in yeast. Nucleic Acids Res. 43: 7306–14.